# Understanding Adversarial Robustness of Symmetric Networks

**Sandesh Kamath**
Chennai Mathematical Institute
ksandeshk@cmi.ac.in

**Amit Deshpande**
Microsoft Research
amitdesh@microsoft.com

## Abstract

Neural network-based models for vision are known to be vulnerable to various adversarial attacks. Some adversarial perturbations are model-dependent, and exploit the loss function gradients of the models to make very small, pixel-wise changes. Other adversarial perturbations are model-agnostic, and include spatial transformations such as rotations, translations, scaling etc. Convolutional Neural Networks (CNNs) are translation equivariant by construction but recent work by Engstrom et al. (2017) has shown that they too are vulnerable to natural adversarial attacks based on rotation and translation.

In this paper, we consider Group-equivariant Convolutional Neural Networks (GCNNs) proposed by Cohen & Welling (2016) that are rotation equivariant by construction, and study their robustness to adversarial attacks based on rotations as well as pixel-wise perturbations. We observe that GCNNs are robust to small degrees of rotations away from the ones present in the training data. We also observe that applying data augmentation increases their robustness.

## 1 Introduction

Neural network-based models achieve state of the art results on several speech and visual recognition tasks but these models are known to be vulnerable to various adversarial attacks. Szegedy et al. (2013) show that small, pixel-wise changes that are almost imperceptible to the human eye can make neural networks models grossly misclassify. They find a small perturbation so as to maximizes the prediction error of a given model using box-constrained L-BFGS. Goodfellow et al. (2015) propose the Fast Gradient Sign Method (FGSM) as a faster approach to find such an adversarial perturbation given by $x' = x + \epsilon \, \text{sign} \left( \nabla_x J(\theta, x, y) \right)$, where $x$ is the input, $y$ represents the targets, $\theta$ represents the model parameters, and $J(\theta, x, y)$ is the cost used to train the network.

Subsequent work has introduced multi-step variants of FGSM, notably, an iterative method by Kurakin et al. (2017) and Projected Gradient Descent (PGD) by Madry et al. (2018). On visual tasks, the adversarial perturbation must come from a set of images that are perceptually similar to a given image. Goodfellow et al. (2015) and Madry et al. (2018) study adversarial perturbations from the $\ell_\infty$-ball around the input $x$, namely, each pixel value is perturbed by a quantity within $[-\epsilon, +\epsilon]$. Broadly, all the above-mentioned adversarial attacks are model-dependent. Tramer et al. (2017) also mention model-agnostic perturbations using the direction of the difference between the intra-class means.

There is a large class of spatial transformations including translations, rotations, scaling that preserve perceptual similarity. Convolutional Neural Networks (CNNs) are translation-equivariant by construction. However, Engstrom et al. (2017) show that simple adversarial attacks using rotations and translations can fool CNNs, even when they are adversarially trained to make them robust to $\ell_p$-bounded adversaries. They observe that $\ell_p$-bounded and spatial adversarial perturbations have additive or super-additive effect on the performance drop, suggesting that these two types of attacks have no bearing on each other. Engstrom et al. (2017) also show that CNNs achieve translation invariance only if the training data (or augmentation) contains some amount of translated inputs, however, their accuracy against the worst-case translations is significantly worse than the average-case.

CNNs are translation-equivariant but not equivariant with respect to other spatial symmetries such as rotations, reflections etc. Variants of CNNs to achieve rotation-equivariance and other symmetries have received much attention recently, notably, Harmonic Networks (H-Nets) by Worrall et al. (2016), cyclic slicing and pooling by Dieleman et al. (2016), Tranformation-Invariant Pooling (TI-Pooling) by Laptev et al. (2016), Group-equivariant Convolutional Neural Networks (GCNNs) by Cohen & Welling (2016), Steerable CNNs by Cohen & Welling (2017), Deep Rotation Equivariant Networks (DREN) by Li et al. (2017), Rotation Equivariant Vector Field Networks (RotEqNet) by Marcos et al. (2017), Polar Transformer Networks (PTN) by Esteves et al. (2018).

For our study, we choose GCNNs as they achieve close to the current state of the art results on MNIST-rot[1] and CIFAR10 data sets as reported in Esteves et al. (2018). GCNNs provide good representative networks to understand the effect of $\ell_p$-bounded and spatial transformation adversaries on symmetry networks. GCNNs use G-convolutions, they have more weight-sharing than regular convolution layers, and they are easy to implement with minimal computational overhead for discrete groups of symmetry generated by translations, reflections, and rotations. In our implementation $\ell_p$-bounded adversaries are obtained by FGSM algorithm.

## 2    SUMMARY OF OUR RESULTS

We study the robustness of GCNNs to adversarial attacks based on rotations as well as pixel-wise perturbations on MNIST and CIFAR10 data sets, in comparison with Standard CNNs(StdCNNs). The main takeaways of our empirical results are (a) GCNNs are robust to small degrees of rotations away from the ones present in the training data, (b) applying data augmentation increases their robustness, (c) GCNNs achieve state of the art results with smaller sample size.

For Figure (1)(left), we train the networks normally and use test time augmentation. In these experiments, no FGSM perturbations were added. We vary the range from which random rotations are added to *test samples*. We stop at $40°$ as accuracy of both the networks have degraded around 5-10% from the unperturbed test samples. This experiment shows how much the networks are able to generalize, as the MNIST training data does not have any large rotations, only the natural, small rotations. The plot clearly shows that GCNNs outperform StdCNNs, and are also robust to small angles of rotations as their accuracy is well above 98% for rotations upto $20°$.

For Figure (1)(right), we compare the performance of the networks with changing training sample size with and without FGSM adversarial training. In these experiments, we do not augment train/test with rotations. We notice that adversarial training of MNIST helps in achieving better accuracy and also they achieve their best performance safely within 10*k* - 30*k* training sample. This confirms that GCNNs by exploiting symmetry do help reducing training sample size.

For plots in Figure (2), we augment both the train and test with random rotations from the same fixed range to handle spatial perturbations, and see that GCNNs not only out perform StdCNNs but also maintain accuracy above 98% even for rotations in the larger range of $180°$. With this setup, we perform 3 types of experiments using FGSM perturbations: (1) Only adversarial training, i.e., networks are retrained with adversaries, (2) only adversarial testing, i.e., networks are trained normally but tested against adversarial test points, and (3) adversarial training and testing. We observe a similar trend as Figure (1)(right) even for data with varying ranges of rotations. GCNNs with and without adversarial training outperform StdCNNs. In the adversarial test case, our observation strengthens GCNNs case that it's still robust to FGSM attacks to small rotations. Finally, adversarially training of GCNNs does make them more robust to adversarial attacks of rotated test samples.

We perform the same set of above experiments for the CIFAR10 dataset. However, we only report results for the third experiment as for the other two experiments we can see from Figure (3) that their performance is below 50%. We allow perturbation upto $\epsilon = 0.06$ in each coordinate of the image. With adversarial training and adversarial testing the accuracy of both the networks improves significantly.

---

[1]http://www.iro.umontreal.ca/ lisa/twiki/bin/view.cgi/Public/MnistVariations

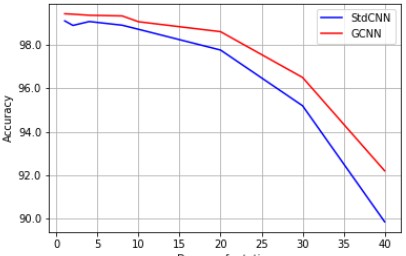 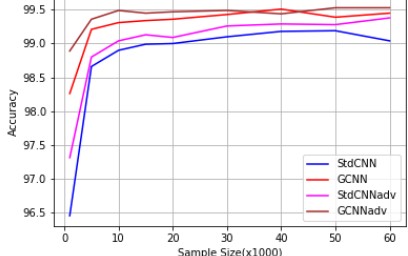

Figure 1: On MNIST, (left) networks trained normally, test augmented with random rotations in $[-x°, x°]$ range. (right) StdCNNs and GCNNs tested against varying training sample size on X-axis, StdCNNAdv and GCNNAdv FGSM-adversarially trained with changing sample size.

We observe similar trends for H-Nets. See Figure (4) in the Appendix.

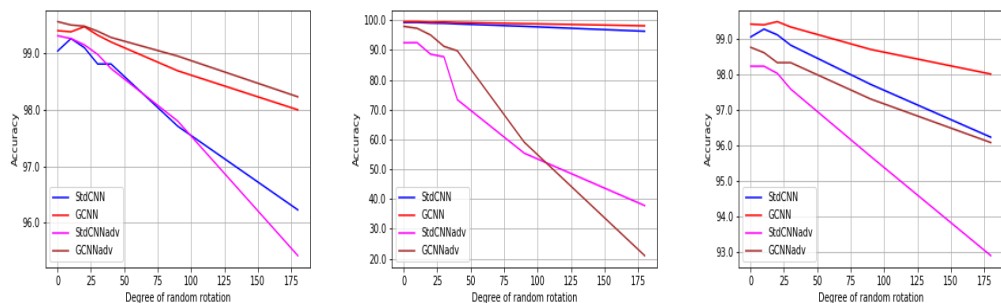

Figure 2: On MNIST, (left) with FGSM adversarial training, (center) without FGSM training and test FGSM perturbed, (right) both train and test FGSM perturbed.

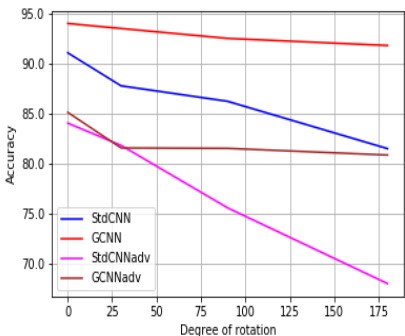

Figure 3: On CIFAR10, (1) StdCNN and GCNN are both trained and tested with random rotation augmentation in the range indicated on X-axis, (2) StdCNNAdv and GCNNAdv indicate FGSM adversarial training and testing with random rotation augmentation in the range indicated on X-axis.

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

## A    DETAILS OF EXPERIMENTS

**Data sets**    MNIST[2] dataset consists of $70,000$ images of $28 \times 28$ size, divided into 10 classes. $55,000$ used for training, $5,000$ for validation and $10,000$ for testing. We use the 7 layer architecture of GCNN similar to Cohen & Welling (2016). CIFAR10[3] dataset consists of $60,000$ images of $32 \times 32$ size, divided into 10 classes. $40,000$ used for training, $10,000$ for validation and $10,000$ for testing. We use the ResNet18 architecture as in He et al. (2016). Input training data are augmented with random cropping and random horizontal flips, apart from the specific augmentations mentioned in Section 2.

**Model architectures**    See Table 1.

**Tables of accuracy on MNIST and CIFAR10 with adversarial training/testing**    See Table 2 and Table 3.

---

[2]http://www.iro.umontreal.ca/ lisa/twiki/bin/view.cgi/Public/MnistVariations
[3]https://www.cs.toronto.edu/ kriz/cifar.html

| Standard CNN | GCNN[4] |
|---|---|
| Conv(10,3,3) + Relu | P4ConvZ2(10,3,3) + Relu |
| Conv(10,3,3) + Relu | P4ConvP4(10,3,3) + Relu |
| Max Pooling(2,2) | Group Spatial Max Pooling(2,2) |
| Conv(20,3,3) + Relu | P4ConvP4(20,3,3) + Relu |
| Conv(20,3,3) + Relu | P4ConvP4(20,3,3) + Relu |
| Max Pooling(2,2) | Group Spatial Max Pooling(2,2) |
| FC(50) + Relu | FC(50) + Relu |
| Dropout(0.5) | Dropout(0.5) |
| FC(10) + Softmax | FC(10) + Softmax |

Table 1: Architectures used for experiments

| Model | Accuracy(%) | Adv Train(%) | Adv Test(%) | Adv (Train+Test)(%) |
|---|---|---|---|---|
| StdCNN | 99.04 | 99.38 | 92.39 | 98.22 |
| GCNN | 99.45 | 99.56 | 97.89 | 98.75 |

Table 2: MNIST - Accuracy of classification without adversary train/test, with adversarial train, with adversarial test and with adversarial train/test

| Model | Accuracy(%) | Adv Train(%) | Adv Test(%) | Adv (Train+Test)(%) |
|---|---|---|---|---|
| StdCNN | 91.11 | 13.33 | 37.87 | 84.08 |
| GCNN | 94.04 | 9.91 | 41.02 | 85.16 |

Table 3: CIFAR10 - Accuracy of classification without adversary train/test, with adversarial train, with adversarial test and with adversarial train/test

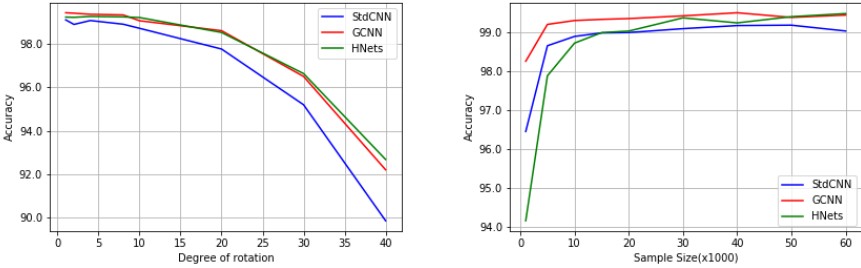

Figure 4: On MNIST, (left) networks trained normally, test augmented with random rotations in $[-x°, x°]$ range. (right) StdCNN, GCNN & H-Nets tested against varying training sample size on X-axis.

