# OpenReview forum: "Understanding Adversarial Robustness of Symmetric Networks"
_ICLR.cc/2018/Workshop — Reject_

### Official Review · AnonReviewer3 · 2018-03-04
**Understanding Adversarial Robustness of Symmetric Networks**

**Rating:** 5
**Confidence:** 4

**Review:**

The robustness of GCNNs to adversarial samples based on pixel-wise perturbations as well as rotation-based adversarial samples is investigated. Several experiments are presented that seem to corroborate the finding that GCNNs are significantly more data-efficient, and possibly more robust to adversarial samples.

It is interesting to study the robustness of various models, including equivariant ones, to geometric and l_p-bounded / pixelwise adversarial attacks. However, the current paper is insufficiently clear about what experiments were performed exactly, and the CNN and GCNN model may not be comparable (see below).

Table 1 shows that the GCNN is obtained from the CNN by replacing Conv with P4ConvP4. As stated by Cohen & Welling in their paper “Group Equivariant Convolutional Networks”, this will increase the number of parameters, unless the number of channels is reduced. To keep the number of parameters the same, the GCNN should have 2x fewer “p4 group channels” than the CNN has planar channels. A GCNN with 2x fewer “p4 group channels” has 2x *more* planar channels, (because 4 rotations are applied to each filter), while retaining the same number of parameters.

Given that the number of parameters is different, it is hard to interpret most of the results in this paper. I would suggest re-running everything with comparable models. It is probably a good idea to increase the model size (e.g. width) of both models a bit, because the current models are quite small. Another architectural question to consider is whether the GCNN should be made rotation-invariant (by pooling over rotations), or not (by using an FC layer at the end). This will have implications for the interpretation of the results.

Several things are not clear from the paper:
- The main findings in the first paragraph of section 2 refers to “robustness” but does not say whether these models are robust to adversarial or random rotations.
- “For Figure (1)(left), we train the networks normally and use test time augmentation.” The phrase “test time augmentation” is not clear, as it could refer to either
  1) Testing on randomly rotated test data
  2) Feeding n rotated copies of an input to the model, and averaging predictions (at test time).
My understanding is that option 1) is what was done.
- For figure 1 (right), it is not clear what kind of adversarial training is performed. Did you use rotation-based adversarial samples, or pixel-wise adversarial samples? The introduction mentions both: “GCNNs provide good representative networks to understand the effect of l_p-bounded and spatial transformation adversaries on symmetry networks.”. From the rest of the paper I would guess that only l_p-bounded (i.e. pixel-wise) adversarial training was used in the experiments.
- “In the adversarial test case, our observation strengthens GCNNs case that it’s still robust to FGSM attacks to small rotations". I don’t understand this statement. GCNNs were not designed to withstand pixel-wise adversarial attacks, so how is their case strengthened? (The robustness to random rotation does strengthen the case for GCNNs, of course)
- In figure 2 (middle), how are the “CNN” and “GCNN” models trained and tested? The middle plot is labelled “without FGSM training and test FGSM perturbed”, so one would think that all 4 lines correspond to test-time FGSM-perturbed results. But then it’s not clear what the difference is between StdCNN and StdCNNAdv or GCNN and GCNNAdv. I guess CNN and GCNN are not adversarially perturbed at all?
- Is it really fair to interpret the GCNNAdv vs CNNAdv results in fig 2 (middle) as showing that GCNNs are more robust to adversarial samples? I’m not sure this is what the paper claims, but the difference does not seem to be very large, and given that this is only one MNIST experiment with one architecture, with no error bars, I don’t think it provides sufficient evidence for this claim.

Suggestion: for plot 1 (right), I would use a logarithmic spacing of sample sizes. E.g. train on 10, 100, 1000, 55000 (max), and space each 10x by one unit on the x-axis. This should make it easier to read off approximately how much more data the CNN needs than the GCNN. E.g. GCNN might need 10x less data to reach the same performance. Right now a large part of the graph is used for the range where all models achieve near-maximal performance.

Although this review mainly focussed on the weak points in this paper, I think the paper does have potential. If the missing details are provided and the comparability of CNN/GCNN is ensured, I would be willing to recommend acceptance of this paper.

---

### Official Review · AnonReviewer2 · 2018-03-09

**Rating:** 4
**Confidence:** 3

**Review:**

The paper compares the robustness of ordinary CNNs and Group CNNs (GCNNs) to image rotation and adversarial perturbations. The authors find that (quote from the paper) “(a) GCNNs are robust to small degrees of rotations away from the ones present in the training data, (b) applying data augmentation increases their robustness, (c) GCNNs achieve state of the art results with smaller sample size.”

Pros:
- Analysis of CNNs and their more advanced variants is an important direction of research
- I am not aware of other papers performing experiments similar to those made by the authors

Cons:
- The paper does not make any technical contribution, but concentrates on analysis. This is not a problem by itself, but the analysis then has to be extremely interesting, insightful or surprising.
- However, the claimed contributions seem very expected: advanced network design improves the invariance to rotations and the sample complexity (this is exactly what it was designed for), and it is not a surprise that data augmentation improves the robustness.
- The authors make use of adversarial training, but to my knowledge this technique only allows fighting a specific type of adversarial perturbations used during training. This limits the practical applicability of the method. Thus the relevance of the proposed analysis, based on adversarial training, is questionable.

To conclude, the paper addresses a generally interesting problem, but does not seem to make a very significant contribution. Therefore I tend to recommend rejection.

---

### Official Review · AnonReviewer1 · 2018-03-11
**Random rotations are not adversarial**

**Rating:** 4
**Confidence:** 3

**Review:**

The paper shows that GCNNs are resistant to small rotations away from those in the training data and that applying data augmentation increases their robustness to these rotations.

The paper describes these rotations as adversarial even though they are standard image transformations. The adversarial rotations don't target any specific class. In this sense, I could call adding random noise or darkening an image to be 'adversarial'. The results given are also unsurprising, in particular that the GCNN has better accuracy than a standard CNN when the test data is rotated. The fact that performing data augmentation improves this is also unsurprising.

What might be more interesting is to explain why GCNN performance still drops so much with rotations but this is not attempted.

---

### Decision · Program_Chairs · 2018-03-20
**ICLR 2018 Workshop Acceptance Decision**

**Decision:**

Reject

**Comment:**

Based on the reviews, this paper has not been accepted for presentation at the ICLR workshop. However, the conversation and updates can continue to appear here on OpenReview.